# Influence of Bias Correction Methods on Simulated Köppen−Geiger Climate Zones in Europe

**Beáta Szabó-Takács [1,\*], Aleš Farda [1,2], Petr Skalák [1,2] and Jan Meitner [1]**

[1]  Global Change Research Institute CAS, 60300 Brno, Czech Republic; farda.a@czechglobe.cz (A.F.);
    skalak.p@czechglobe.cz (P.S.); meitner.j@czechglobe.cz (J.M.)
[2]  Czech Hydrometeorological Institute, 143006 Prague, Czech Republic
\*  Correspondence: szabo.b@czechglobe.cz

**Abstract:** Our goal was to investigate the influence of bias correction methods on climate simulations over the European domain. We calculated the Köppen−Geiger climate classification using five individual regional climate models (RCM) of the ENSEMBLES project in the European domain during the period 1961−1990. The simulated precipitation and temperature data were corrected using the European daily high-resolution gridded dataset (E-OBS) observed data by five methods: (i) the empirical quantile mapping of precipitation and temperature, (ii) the quantile mapping of precipitation and temperature based on gamma and Generalized Pareto Distribution of precipitation, (iii) local intensity scaling, (iv) the power transformation of precipitation and (v) the variance scaling of temperature bias corrections. The individual bias correction methods had a significant effect on the climate classification, but the degree of this effect varied among the RCMs. Our results on the performance of bias correction differ from previous results described in the literature where these corrections were implemented over river catchments. We conclude that the effect of bias correction may depend on the region of model domain. These results suggest that distribution free bias correction approaches are the most suitable for large domain sizes such as the pan-European domain.

**Keywords:** Regional Climate Model; climate classification; bias correction methods; precipitation; temperature

## 1. Introduction

Climate classifications are frequently applied tools for evaluating the real climate system. One of the oldest and still widely accepted systems of climate types was introduced by Wladimir Köppen [1] and later modified by Geiger [2] and additionally by Trewartha [3–7]. Köppen divided eleven climate types based on annual and monthly changes in temperature and precipitation. Trewartha modified the Köppen classification so that the classifications based on the main quality differences and the vegetation characteristics were better taken into consideration. The so-called Köppen–Geiger (K-G) climate classification is derived directly from eco-biological vegetation characteristics within the individual regions of the Earth, which make it suitable for assessing climate change impacts on ecosystems. It is based on annual and monthly mean values of temperature and precipitation and distinguishes five main vegetation groups: the equatorial zone (A), the arid zone (B), the warm temperate zone (C), the snow zone (D) and the polar zone (E). The main groups are further divided into subtypes, reflecting the annual course of air temperature or precipitation and their monthly values compared to a defined threshold. For a detailed overview of all K-G classes and their spatial distribution around the world, we refer to [8]. K-G classification can be applied either to the real observed data of the Earth's climate or present or future conditions simulated by climate models [5,7,9,10]. Some studies, e.g., [11,12] have used the Köppen–Trewartha classification [13] to map the extent of climate change in Europe using

an ensemble mean of regional climate models (RCMs) and simulations, considering the uncertainty related to driving global climate models (GCMs). However, the fact remains that all studies based on climate models should deal with model errors carefully before drawing conclusions.

According to [14], model errors can be caused by the initial and boundary conditions, parameterization, physical formulation, internal variability or model shortcomings [15–19]. Model errors can be divided into two categories: unsystematic errors (random) and systematic errors (bias). Random errors stem from the internal variability of climate models, which are a dominant source of uncertainty for shorter (decadal) timescales in model simulations [20]. Bias is defined as any systematic discrepancy of model simulation and observation. Systematic errors can originate either from inadequately constrained parameters or from model structures that are unable to describe the physical process of interest [21]. Model bias is the most prevalent source of uncertainty for longer (century) timescales [20]. Moreover, bias corrected climate model outputs may lead to a significant response in some impact models as decision support tools [22–24].

In our previous work [25] we applied the K-G classification as a diagnostic tool of climate change for six RCM experiments originally produced as a part of the EU FP6 project, ENSEMBLES [26]. Every experiment represented one specific RCM, driven by one of two GCMs. The simulations followed the A1B emission scenario of Intergovernmental Panel on Climate Change (IPCC) [27,28], and the results were evaluated for the near (2021−2050) and far (2071−2100) future periods. The model simulations were subjected to validation and bias correction using the empirical distribution mapping technique on E-OBS [29] observed data as a reference. We found that warmer climate type increased in each RCM for the future but the degree of their extension was different among them. These differences came from the different GCM applications as the driver, the different physical packages of RCMs and the different representations of natural variability in individual models.

Owing to the fact that any choice of bias correction method can be an additional source of uncertainty [23], in this study, we aim to quantify the impacts of different bias correction techniques on the simulated distribution of K-G zones over Europe. The influence of different bias correction methods has been studied over small geographical domains, usually select river basins in Scandinavia [30], North America [31], or North-estern China [32]. In these studies, the performance of bias correction methods was investigated by statistical indices. References [31] and [30] suggested distribution-based methods, while [32] found that the quantile mapping and power transformation of precipitation methods performed equally best in terms of the frequency-based indices, while the local intensity scaling (LOCI) method performed the best in terms of the time-series-based indices. We intend to test the performance of bias correction over a large pan-European domain, as the bias varies in regions of the domain. Moreover, we study the bias correction performance by implementing Köppen–Geiger climate classification.

Our two major research questions are as follows:

Which bias correction methods of precipitation and temperature are able to reproduce climate classification based on the observed parameters in the 1961–1990 time period?

Which bias correction methods of precipitation and temperature are the most reliable for climate prediction over the whole pan-European domain?

This paper is organized as follows: In Section 2, a short description of the K-G classification, selected models and applied bias corrections are presented. In Section 3, the resulting climate classification with respect to the individual bias correction method is presented. Section 4 contains a discussion of our findings and Section 5 offers the conclusions we draw.

## 2. Materials and Methods

### 2.1. K-G Classification

Köppen and Geiger classified climate based on annual and monthly mean values of temperature and precipitation. Table 1 contains the methodology to calculate K-G climate zones in Europe based on [8].

**Table 1.** Key to calculate K-G zones in Europe and their third index. Pann is the accumulated annual precipitation. Pmin is the precipitation of the driest month. Psmin, Psmax, Pwmin and Pwmax are defined as the lowest and highest monthly precipitation values for the summer and winter half-years. Pth is the dryness threshold. Tann is the annual mean temperature, and the monthly mean temperatures of the warmest and coldest months are marked by Tmax and Tmin, respectively. The precipitation and temperature are given in mm and °C, respectively.

| Type | Description | Criterion |
| --- | --- | --- |
| **B** | Arid climates | $P_{ann} < 10\,P_{th}$ |
| BS | Steppe climates | $P_{ann} > 5\,P_{th}$ |
| BW | Desert climates | $P_{ann} \leq 5\,P_{th}$ |
| **C** | Warm temperate climates | $-3\,°C < T_{min} < +18\,°C$ |
| Cs | Warm temperate climates with dry summers | $P_{smin} < P_{wmin}$, $P_{wmax} > 3\,P_{smin}$ and $P_{smin} < 40$ mm |
| Cw | Warm temperate climates with dry winters | $P_{wmin} < P_{smin}$ and $P_{smax} > 10\,P_{wmin}$ |
| Cf | Warm temperate climates, fully humid | neither Cs nor Cw |
| **D** | Snow climates | $T_{min} \leq -3\,°C$ |
| Ds | Snow climates with dry summers | $P_{smin} < P_{wmin}$, $P_{wmax} > 3\,P_{smin}$ and $P_{smin} < 40$ mm |
| Dw | Snow climates with dry winters | $P_{wmin} < P_{smin}$ and $P_{smax} > 10\,P_{wmin}$ |
| Df | Snow climates, fully humid | neither Ds nor Dw |
| **E** | Polar climates | $T_{max} < +10\,°C$ |
| ET | Tundra climates | $0\,°C \leq T_{max} < +10\,°C$ |
| EF | Frost climates | $T_{max} < 0\,°C$ |
| **third index for C and D climate zones** | | |
| Type | Description | Criterion |
| a | Hot summers | $T_{mean} > 22°C$ |
| b | Warm summers | not (a) and at least 4 $T_{mon} \geq +10\,°C$ |
| c | Cool summers and cold winters | not (b) and $T_{min} > -38°C$ |
| d | Extremely continental | like (c) and $T_{min} \leq -38°C$ |
| **third index for B climate zone** | | |
| Type | Description | Criterion |
| h | Hot steppe/desert | $T_{ann} \geq +18\,°C$ |
| k | Cold steppe/desert | $T_{ann} < +18\,°C$ |

The dryness threshold is calculated by

$$P_{th} = \begin{cases} 2\{T_{ann}\} \ if \ at \ least \ \frac{2}{3} of \ the \ annual \ precipitation \ occurs \in winter \\ 2\{T_{ann}\} + 28 \ if \ at \ least \ \frac{2}{3} of \ the \ annual \ precipitation \ occurs \in summer \\ 2\{T_{ann}\} + 14 \ otherwise \end{cases}.$$

## 2.2. Datasets and Bias Corrections

For the analysis of bias correction influence on K-G zone distribution in Europe, we used simulations of five regional climate models from the ENSEMBLES project, as summarized in Table 2. The large scale forcing for two RCMs was taken from driving ARPÉGE GCM, and three of them were driven by ECHAM5-r3 GCM. The E-OBS version 10 gridded dataset of daily station observations with a spatial resolution of 0.25° in longitude and latitude was used as a reference dataset for validation and bias correction in the period from 1961 to 1990. Before the direct comparison of models and observations, the RCMs were interpolated from their native grids to the E-OBS 0.25 ° regular grid by the nearest neighbour remapping method.

**Table 2.** The institute, global climate models (GCMs), regional climate models (RCMs) and resolution of chosen models from the ENSEMBLES EU project.

|  | INSTITUTE/ REFERENCE | GCM | RCM | RESOLUTION |
|---|---|---|---|---|
| 1 | Centre National de Researches Météorologiques (CNRM)/ [33] | ARPÈGE | ALADIN | 25 km |
| 2 | Danish Meteorological Institute (DMI)/ [34] | ARPÈGE | HIRHAM | 25 km |
| 3 | Koninklijk Nederlands Meteorologisch Instituut (KNMI)/ [35] | ECHAM5-r3 | RACMO2 | 25 km |
| 4 | Swedish Meteorological and Hydrological Institute (SMHI)/ [36] | ECHAM5-r3 | RCA | 25 km |
| 5 | International Centre for Theoretical Physics (ICTP)/ [37] | ECHAM5-r3 | RegCM | 25 km |

The simulated climate zones were analysed in the Alps (AL), the British Isles (BI), Eastern Europe (EA), France (FR), the Iberian Peninsula (IP), the Mediterranean (MD), Mid-Europe (ME) and Scandinavian (SC) regions (Figure 1) specified in the framework of the PRUDENCE project [38].

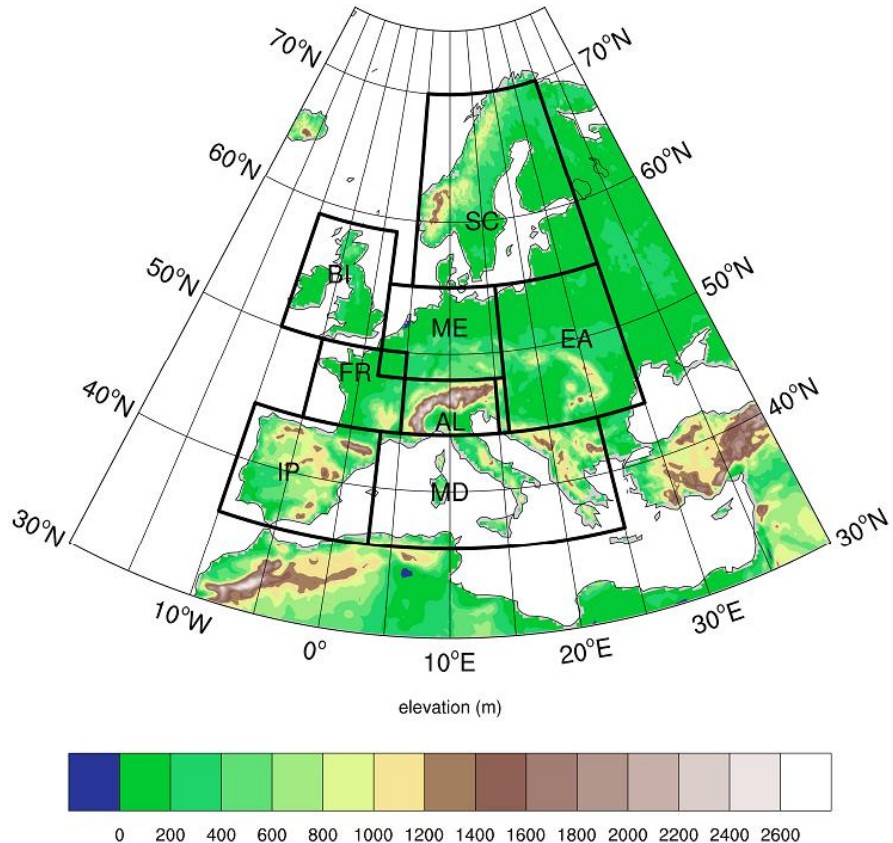

**Figure 1.** Subdomains based on the Prudence project: the Alps (AL), the British Isles (BI), Eastern Europe (EA), France (FR), the Iberian Peninsula (IP), the Mediterranean (MD), Mid-Europe (ME) and Scandinavia (SC).

Figure 2 demonstrates the simulated K-G zones without bias correction. Large differences can be seen between the simulated zones and between the simulated and observed zones. The distribution of K-G zones varied from the K-G zones based on the observed parameters in the case of ARPÈGE driven RCMs. HIRHAM RCM produced dryer climate zones in each region owing to the underestimated precipitation. It produced Csa and Csb zones instead of Cfb in France, Mid-Europe, Eastern Europe and in the Mediterranean and Dsb instead of Dfb in the Alps. Furthermore, the extension of BSk was extremely large in the Iberian Peninsula and in Eastern Europe. Both HIRHAM and ALADIN overestimated the Tundra climate (ET) zone in Scandinavia. In ALADIN simulation the Cfb zone was overestimated in the Iberian Peninsula and in the Italian Peninsula, while it was underestimated in Eastern Europe, in the Mediterranean and on the Western coast of France. The ECHAM5-r3 forced RCMs produced better K-G simulations but the RegCM simulated a wetter climate in the Iberian Peninsula and the Mediterranean, whilst RACMO2 and RCA produced drier climate zones in the Mediterranean and Eastern Europe. Each of them overestimated the ET zone in the Alps.

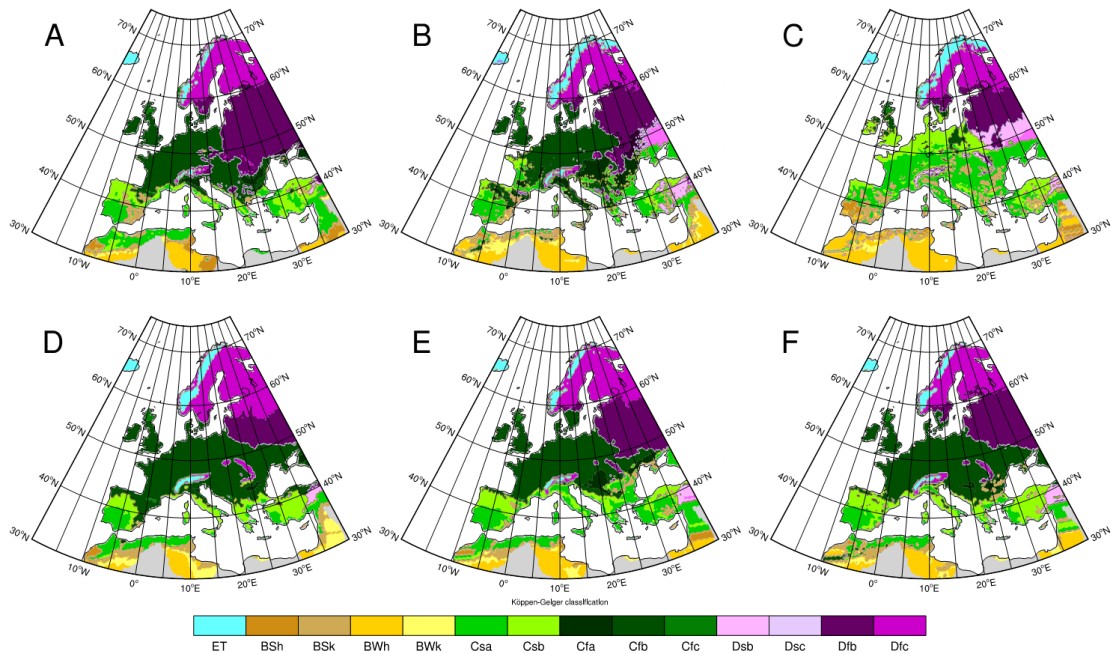

**Figure 2.** Simulated K-G climate classification according to E-OBS (**A**) and in ALADIN (**B**), HIRHAM (**C**), RegCM (**D**), RAMCMO2 (**E**) and RCA (**F**) without bias correction.

We applied the following bias correction methods: i) the empirical quantile mapping (eQM) of precipitation and temperature [15], ii) quantile mapping of precipitation and temperature based on a gamma + Generalized Pareto Distribution (gpQM) [39], iii) the power transformation of precipitation [40,41] the variance scaling of temperature [23], and iv) the local intensity scaling (LOCI) [42].

The daily mean precipitation and temperature values were used for the bias corrections.

### 2.2.1. Empirical Quantile Mapping

Empirical quantile mapping correction was used for correcting the nonparametric empirical cumulative distribution function in simulated daily data. This method calibrates the simulated Cumulative Distribution Function (CDF) by adding both the mean delta change and the individual delta changes in the corresponding quantiles to the observed quantiles. The implemented eQM obtained the correction function for 99 percentiles of observed and simulated distribution and linearly interpolated between two percentiles [15]. Outside the range of percentiles, e.g., for the 99th percentile, a constant correction was applied. In the case of precipitation, a 1 mm threshold value was considered so that the precipitation was redefined to zero if the value was less than 1 mm. We applied this bias correction with a 90-day moving window.

### 2.2.2. Quantile Mapping Based on Gamma + Generalized Pareto Distribution

Quantile mapping based on gamma + generalized Pareto distribution is a quantile mapping method similar to eQM but assumes that the observed and simulated precipitation density distribution are correctly approximated by gamma, and the temperature density distribution is correctly approximated by Gaussian distribution. Therefore, it uses theoretical distribution in the quantile mapping instead of empirical distribution. Due to the fact that gamma distribution is a light-tailed distribution, it is combined with a general Pareto distribution [39]. The observed and simulated quantiles were interpolated by inverse distance weighting. The 1 mm threshold cut off was also applied to precipitation in this approach. The gpQM bias correction with a 90-day moving window in the case of precipitation was used. Owing to the fact that the seasonal temperature density distribution cannot be approximated by Gaussian distribution in some places in Europe [43], the temperature gpQM

correction produces a large number of infinitive values with a 90-day moving window. Therefore, gpQM was applied only in the case of precipitation.

### 2.2.3. Power Transformation of Precipitation

Power transformation of precipitation can be used for adjusting the variance statistics of precipitation. Simulated monthly precipitation is powered by a "b" value that guarantees the coefficient of variance (CV) of the simulated daily precipitation matches the CV of the observed daily precipitation. This power "b" value is estimated on a monthly basis using a 90-day window centred on the interval with a root-finding algorithm. Thereafter, the powered precipitation series is multiplied by the standard linear scaling parameter, which was calculated by dividing the monthly mean observed precipitation by the monthly mean powered simulated precipitation.

### 2.2.4. Variance Scaling of Temperature

Correspondingly, variance scaling of temperature corrects both the mean and variance values of temperature. In the first step, the temperature mean was corrected with the difference between the observed and simulated climatological monthly means. After that, the mean-corrected simulated temperature was shifted on a monthly basis to the zero mean. Thereafter, the standard deviation of the shifted temperature was scaled based on the ratio of the climatological monthly standard deviation of the observed and simulated data. Finally, the standard deviation corrected time series were shifted back using the corrected mean.

### 2.2.5. Local Intensity Scaling of Precipitation

The local intensity scaling correction corrects the mean as well as both the wet-day frequencies and wet-day intensities of precipitation. The frequency of wet-days in the case of observation considers those days when the precipitation value is higher than the 1 mm threshold.

The model's wet-day threshold was determined from the daily RCM precipitation series such that the threshold exceedance matched the wet-day frequency in the observed series. The scaling factor of this correction was calculated based on the ratio of the climatological monthly mean wet-day intensities between the observations and the RCMs with the adjusted wet-day thresholds. Subsequently, the simulated monthly precipitation values were adjusted with the model's wet-day threshold and multiplied by the scaling factor. Finally, the daily simulated precipitation was downscaled from the calibrated monthly scale such that the precipitation values were redefined to zero on those days when the observed precipitation was less than 1 mm.

The equations of bias correction methods are detailed in [44]. Bias corrections and K-G classification were implemented in Matlab by the MeteoLab [45] and Weaclim [46] toolboxes. The figures were created by NCAR Command Language [47].

## 3. Results

### 3.1. Empirical Quantile Mapping with a 90-Day Moving Window

The application of eQM bias correction with a 90-day moving window improved the climate classification. RCMs simulated appropriate climate zones in each region with the exception of HIRHAM (Figure 3). In the case of HIRHAM RCM, the climate zone simulation was improved in the Northern regions, e.g, in Scandinavia, the British Isles and Mid-Europe, but it still produced dryer climate zones in the Iberian Peninsula, the Mediterranean and Eastern Europe. In the other RCMs, the extension of climate zones differed from the observed ones mainly in the Iberia Peninsula, in the Alps, in the Mediterranean and in Eastern Europe. The difference in the frequency of the occurrence of the climate zones was only 1–2% between the RCMs with exception of HIRHAM in each region. For example, the occurrence of the Cfb zone in the Alps was 53%, 51%, 50% and 50% according to the ALADIN, RegCM, RACMO2 and RCA simulations, respectively.

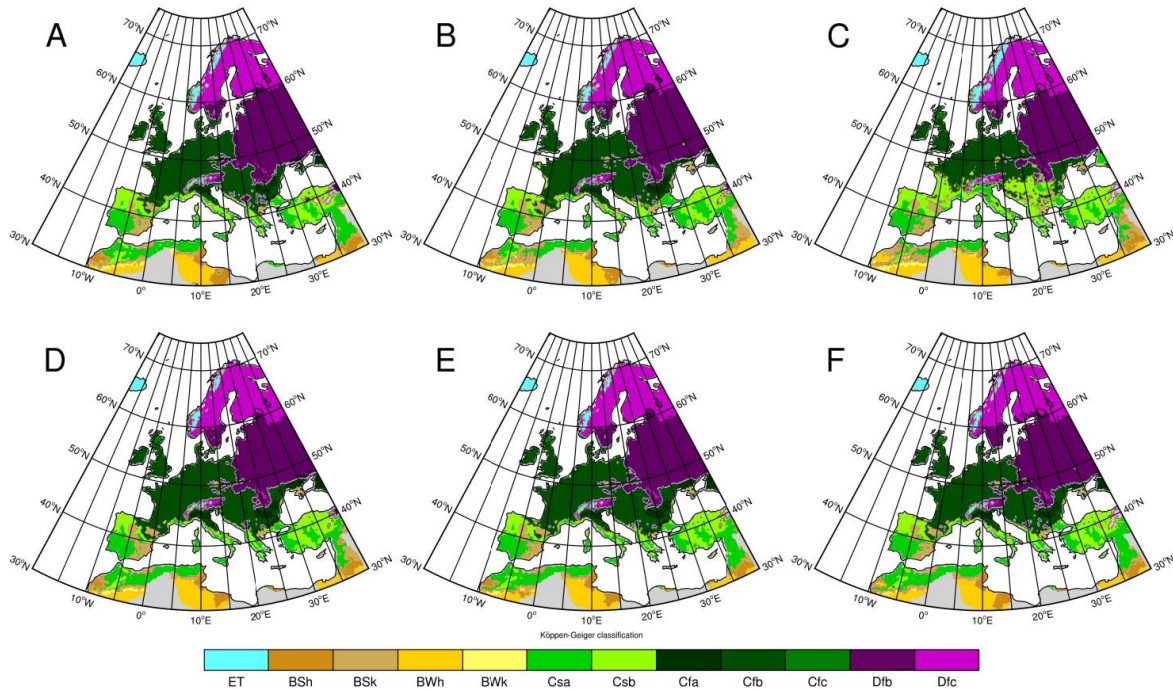

**Figure 3.** Simulated K-G climate classification according to E-OBS (**A**) and empirical quantile mapping (eQM) corrected precipitation and temperature with 90-day moving window in ALADIN (**B**), HIRHAM (**C**), RegCM (**D**), RAMCMO2 (**E**) and RCA (**F**).

The precipitation was mainly underestimated in each season except in the British Isles in winter (DJF) (Table 3a). The eQM with a 90-day moving window decreased the residual temperature bias in DJF but increased it in summer (JJA) in ALADIN, HIRHAM and RegCM in some regions (Table 3b).

**Table 3.** Residual bias of seasonal amount of simulated precipitation (a) and of seasonal mean of simulated temperature (b) in the case of eQM bias correction in eight different regions: the Alps (AL), the British Isles (BI), Eastern Europe (EA), France (FR), the Iberia Peninsula (IP), the Mediterranean (MD), Mid-Europe (ME) and Scandinavia (SC) in DJF and JJA The bias values are in % and in °C in the case of precipitation and temperature, respectively.

| a) | ALADIN | | HIRHAM | | RegCM | | RACMO2 | | RCA | |
|---|---|---|---|---|---|---|---|---|---|---|
| Region | DJF | JJA | DJF | JJA | DJF | JJA | DJF | JJA | DJF | JJA |
| AL | −2 | −9 | 2 | −23 | −4 | −6 | −5 | −6 | −5 | −7 |
| BI | 1 | −2 | 2 | −7 | 1 | −3 | 0 | −3 | 1 | −2 |
| EA | −8 | −11 | −4 | −17 | −7 | −6 | −8 | −3 | −9 | −3 |
| FR | −1 | −5 | 0 | −20 | −6 | 0 | −6 | −5 | −7 | −4 |
| IP | −6 | −5 | −8 | −23 | −8 | −11 | −8 | −21 | −9 | −23 |
| MD | −6 | −8 | 0 | −34 | −4 | −9 | −4 | −9 | −5 | −14 |
| ME | −4 | −9 | −2 | −12 | −4 | −3 | −5 | −3 | −5 | −2 |
| SC | −6 | −1 | −3 | −8 | −3 | −2 | −2 | −2 | −4 | 1 |

−100  −50  −25  −10  10  25  50  100

**Table 3.** *Cont.*

| b) | ALADIN | | HIRHAM | | RegCM | | RACMO2 | | RCA | |
|---|---|---|---|---|---|---|---|---|---|---|
| Region | DJF | JJA | DJF | JJA | DJF | JJA | DJF | JJA | DJF | JJA |
| AL | 0.2 | 0.1 | 0.3 | −0.4 | 0.3 | 0.1 | 0.1 | 0.1 | −0.1 | 0.0 |
| BI | 0.3 | 0.0 | 0.2 | −0.3 | 0.2 | 0.0 | 0.1 | 0.1 | 0.1 | 0.1 |
| EA | 0.5 | −0.2 | 0.4 | −0.8 | 0.4 | 0.0 | 0.2 | 0.0 | 0.0 | 0.0 |
| FR | 0.3 | 0.0 | 0.2 | −0.3 | 0.2 | 0.2 | 0.0 | 0.3 | 0.0 | 0.1 |
| IP | 0.1 | 0.5 | −0.1 | 0.3 | 0.1 | 0.6 | 0.0 | 0.3 | 0.0 | 0.1 |
| MD | 0.1 | 0.6 | 0.1 | −0.3 | 0.2 | 0.4 | 0.0 | 0.4 | −0.1 | 0.2 |
| ME | 0.5 | −0.2 | 0.5 | −0.7 | 0.3 | −0.1 | 0.2 | 0.0 | 0.0 | 0.0 |
| SC | 0.1 | −0.6 | 0.3 | −0.8 | 0.2 | 0.0 | 0.2 | 0.0 | −0.1 | 0.0 |

5   2   1   0.5 −0.5   −1   −2   −5

## 3.2. Quantile Mapping Based on a Gamma + Generalized Pareto Distribution with a 90-Day Moving Window

Owing to the fact the seasonal temperature probability distribution does not fit a Gaussian distribution due to non-Gaussian tails occurrence, the gpQM bias correction with a 90-day moving window was implemented only on the precipitation data. The gpQM with a 90-day moving window bias-corrected precipitation was combined with eQM with 90-day moving window corrected temperature values for the calculation of K-G zones. The gpQM correction with a moving window also improved the climate classification, but it resulted in dryer climate zones in some regions compared to the eQM correction (Figure 4). The Csb and BSk ratio was larger in the Mediterranean and Eastern Europe, respectively, according to the gpQM in each RCM. Owing to gpQM method the extension of dry zones (Csa, Csb, Dsb), the Csb zone was predominant in France, Mid-Europe and the Mediterranean, and the BSk was overestimated in the Iberian Peninsula and Eastern Europe in HIRHAM model.

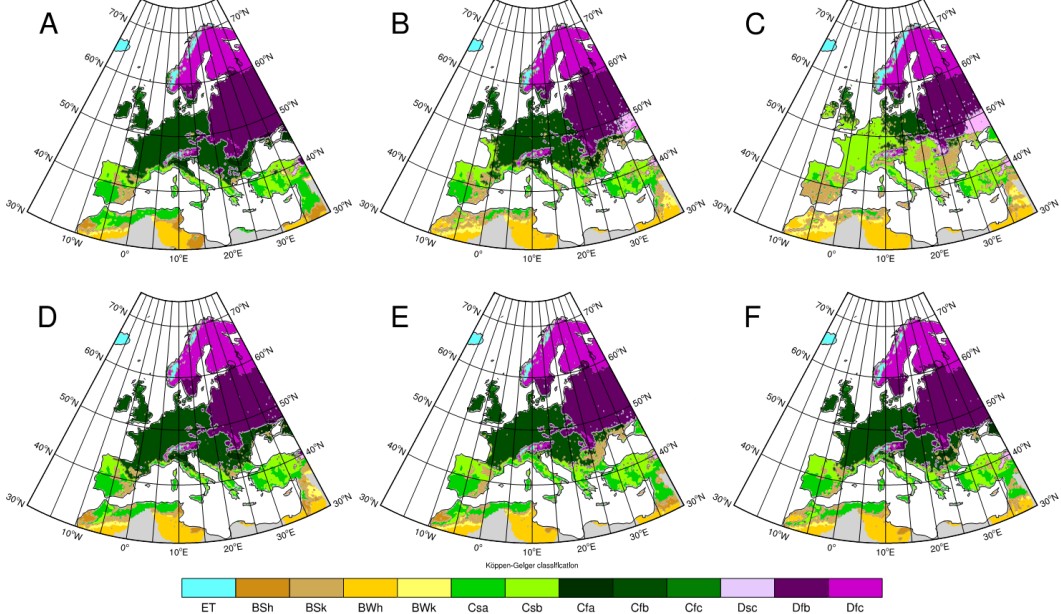

**Figure 4.** Simulated K-G climate classification according to E-OBS (**A**) and quantile mapping of precipitation and temperature based on a gamma + Generalized Pareto Distribution (gpQM) correction of precipitation and eQM correction of temperature with 90-day moving window in ALADIN (**B**), HIRHAM (**C**), RegCM (**D**), RAMCMO2 (**E**) and RCA (**F**).

The residual precipitation bias was variable. The precipitation was overestimated in some regions, mainly in DJF. Although eQM correction resulted in better K-G classification, the residual bias of gpQM correction was smaller in some regions (e.g., in the Mediterranean in the case of RegCM) (Table 4).

**Table 4.** Residual bias in the seasonal amount of simulated precipitation (a) in the case of gpQM bias correction in eight different regions: the Alps (AL), the British Isles (BI), Eastern Europe (EA), France (FR), the Iberian Peninsula (IP), the Mediterranean (MD), Mid-Europe (ME) and Scandinavia (SC) in DJF and JJA. The bias values are in %.

| | ALADIN | | HIRHAM | | RegCM | | RACMO2 | | RCA | |
|---|---|---|---|---|---|---|---|---|---|---|
| **Region** | **DJF** | **JJA** | **DJF** | **JJA** | **DJF** | **JJA** | **DJF** | **JJA** | **DJF** | **JJA** |
| AL | 3 | −12 | 0 | −57 | 5 | −11 | 0 | −22 | 3 | −10 |
| BI | −2 | −5 | −3 | −39 | 3 | −2 | 3 | −4 | 5 | 1 |
| EA | 3 | −20 | 6 | −46 | 3 | −11 | −2 | −22 | −3 | −3 |
| FR | 5 | −11 | −6 | −57 | 0 | 4 | −1 | −9 | 1 | −3 |
| IP | −11 | −10 | −33 | −67 | −3 | −11 | −3 | −36 | −5 | −30 |
| MD | −7 | −15 | −14 | −68 | 0 | −15 | −4 | −49 | −3 | −25 |
| ME | 8 | −16 | 8 | −42 | 5 | −4 | 1 | −9 | 1 | 4 |
| SC | −7 | −1 | 3 | −19 | −1 | 1 | 2 | −2 | −3 | 8 |

| −100 | −50 | −2 | −10 | 10 | 25 | 50 | 100 |

*3.3. Power Transformation of Precipitation and Variance Scaling of Temperature*

The power transformation of precipitation has been implemented in smaller domains in Europe, such as the basin of the river Meuse [40] and the mesoscale catchments of Sweden [30], where the precipitation is significant. In our work, the power value of precipitation was calculated with Brent's root-finding algorithm [48]. It is possible that the mean value of precipitation is near zero in the dryer regions. This zero mean value may have caused an invalid value in the coefficient of variation of precipitation that stopped the root-finding algorithm and produced incorrect K-G zones (this is not shown). To get around this issue, we applied two conditions before running the root-finding algorithm. The first condition was to ignore the RCM precipitation values if they were missing values. The second was to ignore the RCM precipitation values if their mean value was zero, as this causes an invalid value. Thanks to the above-mentioned conditions, the power transformation of precipitation combined with the variance scaling of temperature created the correct K-G classification in each RCM. Negligible differences were seen between the observed and simulated K-G zones (Figure 5). The difference in the frequency of occurrence of climate zones between observations and simulations was zero in each region with the exception of ALADIN. ALADIN simulated larger Cfb and smaller Csb extension in the Iberian Peninsula and in the Mediterranean regions where the difference from the observations was only 2%. Due to these facts, power transformation of precipitation and variance scaling of temperature appear to be the most suitable for climate classification in the whole pan-European domain.

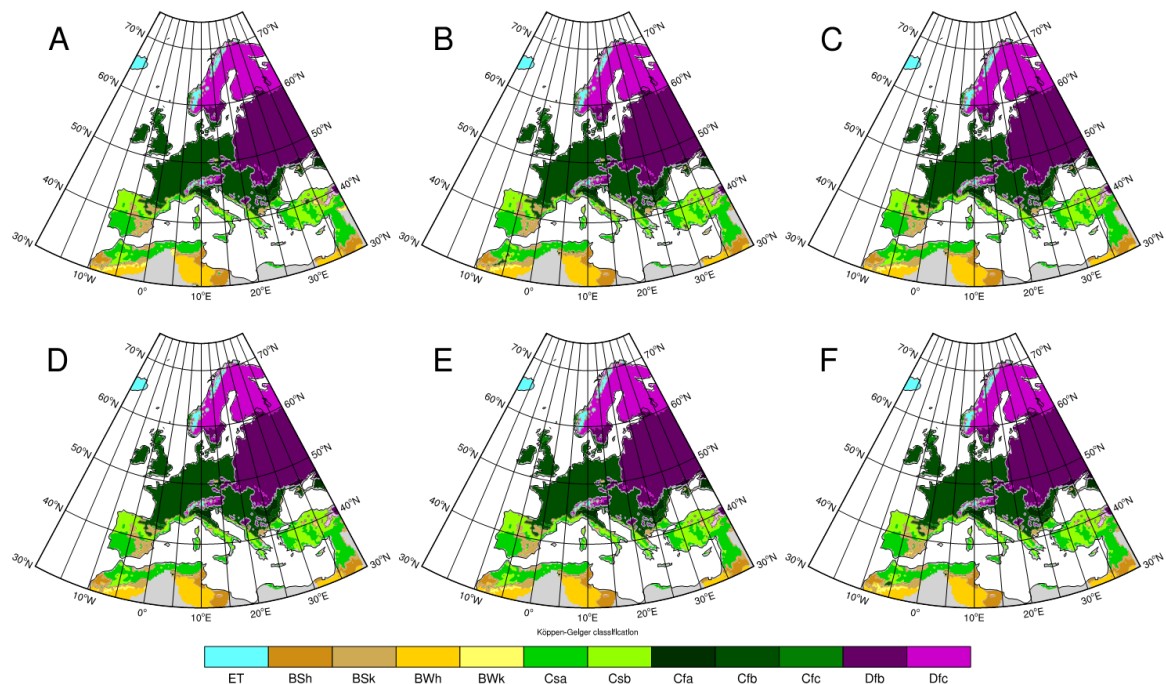

**Figure 5.** Simulated K-G climate classification according to E-OBS (**A**) and power transformation of precipitation and variance scaling of temperature correction in ALADIN (**B**), HIRHAM (**C**), RegCM (**D**), RAMCMO2 (**E**) and RCA (**F**).

The value of residual precipitation bias was similar in each RCM, with the exception of HIRHAM, where the residual bias values were zero (Table 5.). Furthermore, the bias was almost identical, except for HIRHAM, which means that power transformation is not dependent on the RCMs. The modelled temperature was almost commensurate with the observed data when variance scaling correction was implemented.

**Table 5.** Residual bias of seasonal amount of simulated precipitation in the case of power transformation of the precipitation bias correction method in eight different regions: the Alps (AL), the British Isles (BI), Eastern Europe (EA), France (FR), the Iberian Peninsula (IP), the Mediterranean (MD), Mid-Europe (ME) and Scandinavia (SC) in DJF and JJA. The bias values are in %.

| | ALADIN | | HIRHAM | | RegCM | | RACMO2 | | RCA | |
|---|---|---|---|---|---|---|---|---|---|---|
| Region | DJF | JJA | DJF | JJA | DJF | JJA | DJF | JJA | DJF | JJA |
| AL | 0 | −6 | 0 | 0 | 0 | −6 | 0 | −6 | −1 | −6 |
| BI | −8 | 14 | 0 | 0 | −8 | 14 | −8 | 14 | −8 | 14 |
| EA | −9 | −14 | 0 | 0 | −9 | −14 | −9 | −14 | −9 | −14 |
| FR | −5 | 4 | 0 | 0 | −5 | 3 | −5 | 3 | −5 | 4 |
| IP | −12 | 8 | 0 | 0 | −12 | 7 | −12 | 7 | −12 | 7 |
| MD | −10 | 0 | 1 | 0 | −10 | 4 | −10 | 4 | −10 | 4 |
| ME | −9 | −11 | 0 | 0 | −9 | −10 | −9 | −10 | −9 | −10 |
| SC | −12 | 10 | 0 | 0 | −12 | 10 | −12 | 10 | −12 | 9 |

−100    −50    −25    −10    10    25    50    100

### 3.4. Local Intensity Scaling of Precipitation and Variance Scaling of Temperature

Due to the fact that local intensity scaling correction can be applied only to precipitation, it was combined with variance scaling of temperature for the calculation of the K-G classification. Both

corrections are distribution free and correct the diagnostics, as well as the mean. Owing to these facts, the difference between the observed and simulated zones was also negligible, only 1−2% (Figure 6). Apart from that, the RCMs resulted in very similar values.

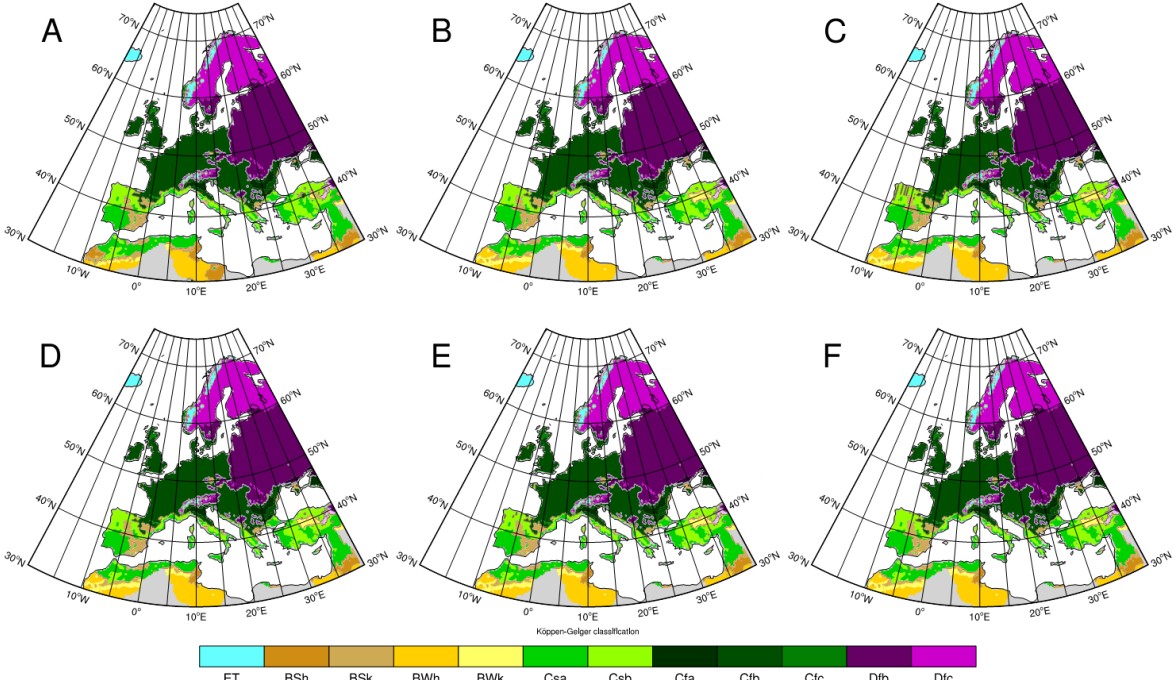

**Figure 6.** Simulated K-G climate classification according to E-OBS (**A**) and local intensity scaling of precipitation and variance scaling of temperature correction in ALADIN (**B**), HIRHAM (**C**), RegCM (**D**), RAMCMO2 (**E**) and RCA (**F**).

In the case of LOCI correction, the residual seasonal precipitation bias was the smallest compared to the other precipitation correction methods, except in the case of the HIRHAM RCM compared to power transformation of the precipitation method (Table 6). This caused a negative bias in both seasons in each RCM. Although the seasonal residual bias values were smaller than in the case of power transformation of precipitation, the minimum and maximum monthly precipitation values of the RCMs were closer to the observed minimum and maximum monthly precipitation values by power transformation in both seasons (not shown). These values determined the subtypes of the K-G zones.

**Table 6.** Residual bias of the seasonal amount of simulated precipitation in the case of local intensity scaling of the precipitation bias correction method in eight different regions: the Alps (AL), the British Isles (BI), Eastern Europe (EA), France (FR), the Iberia Peninsula (IP), the Mediterranean (MD), Mid-Europe (ME) and Scandinavia (SC) in DJF and JJA. The bias values are in %.

| Regio | ALADIN | | HIRHAM | | RegCM | | RACMO2 | | RCA | |
|-------|--------|-----|--------|-----|-------|-----|--------|-----|-----|-----|
| | DJF | JJA | DJF | JJA | DJF | JJA | DJF | JJA | DJF | JJA |
| AL | −1 | −1 | −1 | −1 | −1 | −1 | −1 | −1 | −1 | −1 |
| BI | −2 | −2 | −2 | −2 | −2 | −2 | −2 | −2 | −2 | −2 |
| EA | −6 | −1 | −6 | −2 | −6 | −1 | −6 | −1 | −6 | −2 |
| FR | −2 | −2 | −2 | −2 | −2 | −2 | −2 | −2 | −2 | −2 |
| IP | −1 | −2 | −1 | −2 | −1 | −2 | −1 | −2 | −1 | −2 |
| MD | −2 | −1 | −2 | −1 | −2 | −1 | −2 | 1 | −2 | −1 |
| ME | −5 | −2 | −6 | −2 | −5 | −2 | −6 | −2 | −6 | −2 |
| SC | −5 | −2 | −5 | −2 | −5 | −2 | −5 | −2 | −5 | −2 |

−100  −50  −25  −10  10  25  50  100

### 3.5. Cross-Validation of Bias Corrections

Cross-validation was applied to test bias corrections. Due to the observed annual precipitation and temperature values being nearly stationary (not shown) in the 1961–2000 period, we applied a split-sample test (SST) as advocated by [49]. The parameters were split into calibration and test periods. The bias corrections were calibrated in the first twenty years of 1961–1980, and the corrections were implemented in the second twenty years of 1981–2000. The corrections were validated by the K-G zone simulation in the test period, and the results were compared with the K-G zones based on the observed data in the test period.

### 3.5.1. Validation of Empirical Quantile Correction

Figure 7 shows the simulated K-G distribution based on the validated eQM values of precipitation and temperature compared to K-G zones according to observed parameters in the test period. The ECHAM5-r3 RCMs produced similar results, whilst HIRHAM resulted in a drier climate in Eastern Europe and in the Mediterranean compared to the ALADIN model where the Csb zone was predominant. Each RCM significantly underestimated the BSk zone in the Iberian Peninsula compared to the K-G zones based on observations. In Scandinavia, the ET zone was overestimated with the exception of the ALADIN model, and the Dsc zone expanded in the RACMO2 and RCA models. Moreover, the DSb zone occurred in the RegCM model. In the ECHAM5-r3 driven RCMs, the Dfb zone shifted southwards in Southern Scandinavia. The ratio of the Dfb zone decreased in the Carpathians in Eastern Europe in each RCM. The Cfa zone diminished in the Eastern region of Eastern Europe in the HIRHAM, RACMO2 and RCA models. Moreover, the BSk zone occurred in Eastern Europe in ARPEGE-driven RCMs. The difference between the simulated and observed K-G zones was negligible in Mid-Europe and in the British Isles. The ECHAM5-r3 RCMs simulated a Csb zone in the Western region of France.

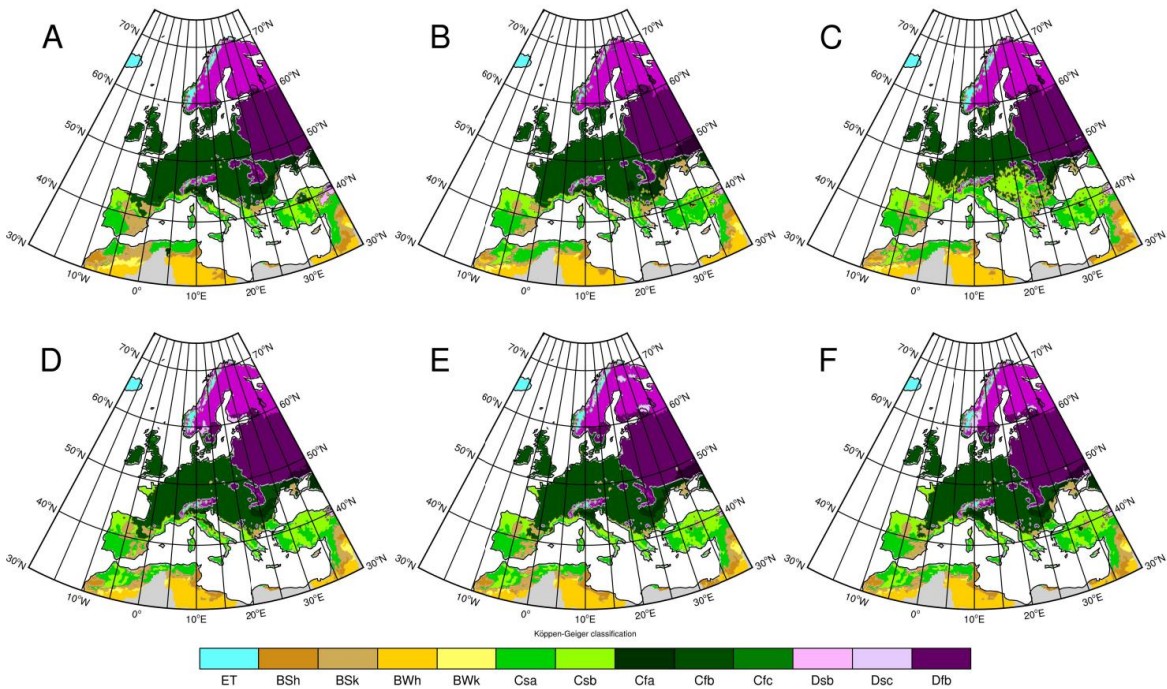

**Figure 7.** Simulated K-G climate classification according to E-OBS (**A**) and eQM corrected precipitation and temperature with 90-day moving window in ALADIN (**B**), HIRHAM (**C**), RegCM (**D**), RAMCMO2 (**E**) and RCA (**F**) in the test period in 1981−2000.

The residual bias of precipitation varied during the season and the eQM correction strongly depended on the regions and the RCMs (Table 7a). Larger residual bias was found in France, in the Iberian Peninsula and in the Mediterranean in each RCM with the exception of ALADIN. The residual bias of temperature was smaller than 1 °C, except in HIRHAM in Scandinavia in the DJF season (Table 7b).

**Table 7.** Residual bias of the seasonal amount of simulated precipitation (a) and of the seasonal mean of the simulated temperature (b) in the case of eQM bias correction in the test period in eight different regions: the Alps (AL), the British Isles (BI), Eastern Europe (EA), France (FR), the Iberia Peninsula (IP), the Mediterranean (MD), Mid-Europe (ME) and Scandinavia (SC) in DJF and JJA. The bias values are in % and in °C for precipitation and temperature, respectively.

| a) | ALADIN | | HIRHAM | | RegCM | | RACMO2 | | RCA | |
|---|---|---|---|---|---|---|---|---|---|---|
| Region | DJF | JJA | DJF | JJA | DJF | JJA | DJF | JJA | DJF | JJA |
| AL | −2 | 3 | 7 | −26 | 3 | −9 | 1 | −2 | 3 | −5 |
| BI | −9 | 7 | −10 | 0 | −12 | −7 | −15 | −9 | −15 | −11 |
| EA | −3 | 5 | 2 | −6 | 4 | 0 | 0 | 5 | 3 | 3 |
| FR | −8 | 4 | −6 | −24 | −17 | −13 | −17 | −19 | −18 | −17 |
| IP | 9 | 3 | 5 | −17 | 13 | −1 | 14 | −18 | 11 | −21 |
| MD | 0 | −5 | 7 | −23 | 14 | 0 | 18 | 12 | 12 | 3 |
| ME | −7 | 10 | −5 | −4 | −7 | −2 | −14 | −2 | −8 | −1 |
| SC | −18 | 0 | −15 | −8 | −14 | −9 | −11 | −11 | −13 | −9 |

−100　−50　−25　−10　　10　　25　　50　　100

**Table 7.** *Cont.*

| b) | ALADIN | | HIRHAM | | RegCM | | RACMO2 | | RCA | |
|---|---|---|---|---|---|---|---|---|---|---|
| Region | DJF | JJA | DJF | JJA | DJF | JJA | DJF | JJA | DJF | JJA |
| AL | 0.3 | −0.4 | 0.3 | −0.4 | −0.2 | −0.4 | −0.5 | −0.5 | −0.6 | −0.5 |
| BI | 0.5 | −0.5 | 0.3 | −0.7 | −0.3 | −0.3 | −0.4 | −0.2 | −0.5 | −0.4 |
| EA | 0.1 | 0.1 | 0.1 | 0.0 | 0.0 | −0.5 | −0.1 | −0.6 | −0.2 | −0.6 |
| FR | 0.2 | −0.9 | 0.2 | −0.9 | −0.6 | −0.4 | −0.7 | −0.3 | −0.9 | −0.5 |
| IP | −0.3 | −0.2 | −0.3 | −0.4 | −0.5 | 0.1 | −0.5 | 0.1 | −0.6 | 0.1 |
| MD | 0.2 | 0.2 | 0.2 | −0.2 | 0.1 | −0.5 | −0.1 | −0.7 | −0.2 | −0.6 |
| ME | 0.2 | −0.3 | 0.3 | −0.5 | −0.2 | −0.5 | −0.3 | −0.6 | −0.4 | −0.7 |
| SC | 0.9 | 0.4 | 1.1 | −0.1 | −0.1 | −0.3 | 0.0 | −0.4 | −0.1 | −0.2 |

| | | | | | | |
|---|---|---|---|---|---|---|
| 5 | 2 | 1 | 0.5 | −0.5 | −1 | −2 | −5 |

### 3.5.2. Validation of Quantile Mapping Based on a Gamma + Generalized Pareto Distribution of Precipitation

Figure 8. demonstrates the simulated K-G zones according to the gpQM of precipitation and eQM of temperature combination during the test period. The gpQM of precipitation resulted in dryer climate zones compare to eQM in Mediterranean region, and a BSk zone was produced in the Southeastern area of Easter-Europe in each RCM except RegCM (Figure 8). The BSK zone dominantly decreased in the Iberian Peninsula in each RCM except HIRHAM. Moreover, significant extension of Csb was simulated in the HIRHAM model in France, Mid-Europe, EasternEurope, in the Mediterranean and in the Southwestern area of the British Isles. In this model, a larger area was covered by the BSk zone than in the other models in the Iberian Peninsula and in Eastern Europe.

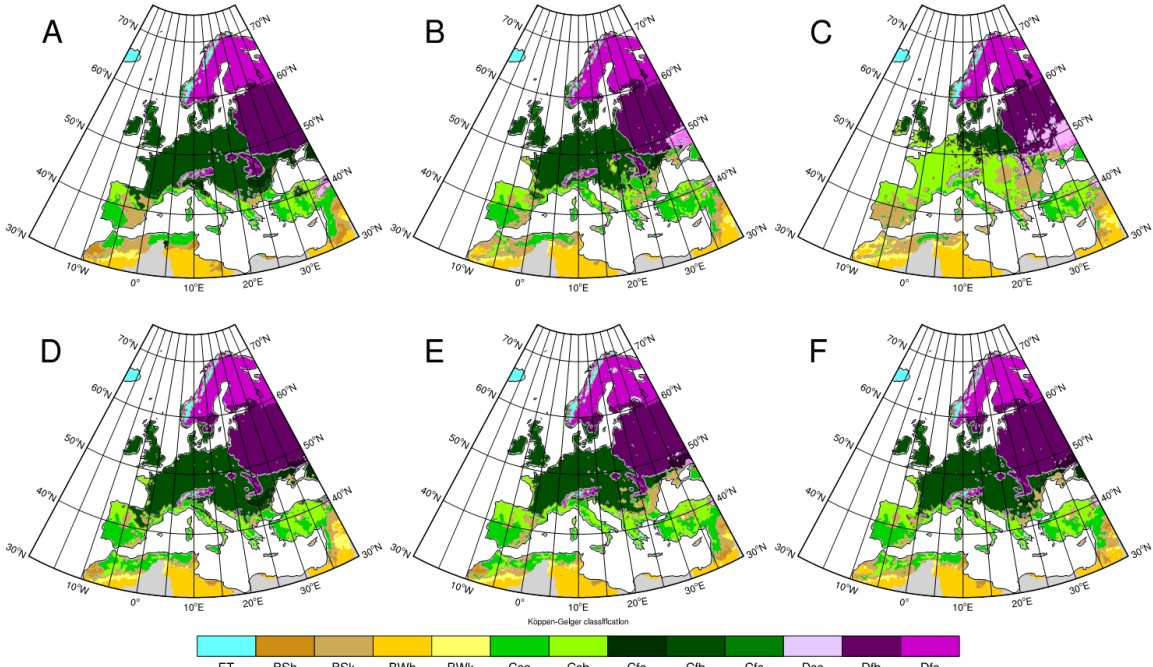

**Figure 8.** Simulated K-G climate classification according to E-OBS (**A**) and gpQM corrected precipitation and eQM corrected temperature with a 90-day moving window in ALADIN (**B**), HIRHAM (**C**), RegCM (**D**), RAMCMO2 (**E**) and RCA (**F**) in the test period in 1981−2000.

Although gpQM correction of precipitation resulted in a larger bias in K-G simulation in several regions compared to the eQM correction, the remained bias was smaller in France and in Scandinavia in ECHAM5-r3 forced RCMs (Table 8). Furthermore, gpQM produced a smaller residual bias in the RegCM model in the British Isles as well.

**Table 8.** Residual bias of seasonal amount of simulated precipitation in the case of gpQM bias correction in the test period in eight different regions: the Alps (AL), the British Isles (BI), Eastern Europe (EA), France (FR), the Iberian Peninsula (IP), the Mediterranean (MD), Mid-Europe (ME) and Scandinavia (SC) in DJF and JJA. The bias values are in %.

| | ALADIN | | HIRHAM | | RegCM | | RACMO2 | | RCA | |
|---|---|---|---|---|---|---|---|---|---|---|
| **Region** | **DJF** | **JJA** | **DJF** | **JJA** | **DJF** | **JJA** | **DJF** | **JJA** | **DJF** | **JJA** |
| AL | 6 | 0 | 6 | −59 | 11 | −12 | 10 | −20 | 12 | −8 |
| BI | −10 | 3 | −13 | −36 | −9 | −5 | −11 | −9 | −10 | −6 |
| EA | 12 | −7 | 11 | −40 | 14 | −5 | 7 | −15 | 9 | 4 |
| FR | 0 | −7 | −10 | −62 | −10 | −10 | −11 | −23 | −10 | −17 |
| IP | 7 | −2 | −24 | −64 | 21 | 0 | 20 | −34 | 16 | −33 |
| MD | 1 | −14 | −8 | −62 | 17 | −8 | 16 | −42 | 15 | −16 |
| ME | 7 | 2 | 3 | −38 | 1 | −1 | −8 | −7 | −2 | 6 |
| SC | −19 | 0 | −11 | −19 | −10 | −5 | −7 | −9 | −11 | 0 |

−100  −50  −25  −10  10  25  50  100

### 3.5.3. Validation of Power Transformation of Precipitation and Variance Scaling of Temperature

The power transformation of precipitation and variance scaling of temperature bias corrections resulted in similar K-G zone distributions in each RCM (Figure 9). The extension of K-G zones was different between the RCMs and differed from the observed ones. The ECHAM5-r3 forced RCMs and HIRHAM simulated larger, while ALADIN resulted in a smaller ET fraction in Scandinavia. Moreover, ECHAM5-r3 forced RCMs simulated a large Dsc zone fraction in the Scandinavian mountains. The ratio of the Dfb zone decreased in the Carpathians in Eastern Europe in each RCM. The Cfa zone expanded in Eastern Europe according to ARPÉGE forced RCMs, whilst it decreased in ECHAM5-r3 driven models. In the Western part of France, the Csb zone was simulated with the exception of the ALADIN model. In the Mediterranean and in the Iberian Peninsula, the BSk zone was underestimated in each RCM.

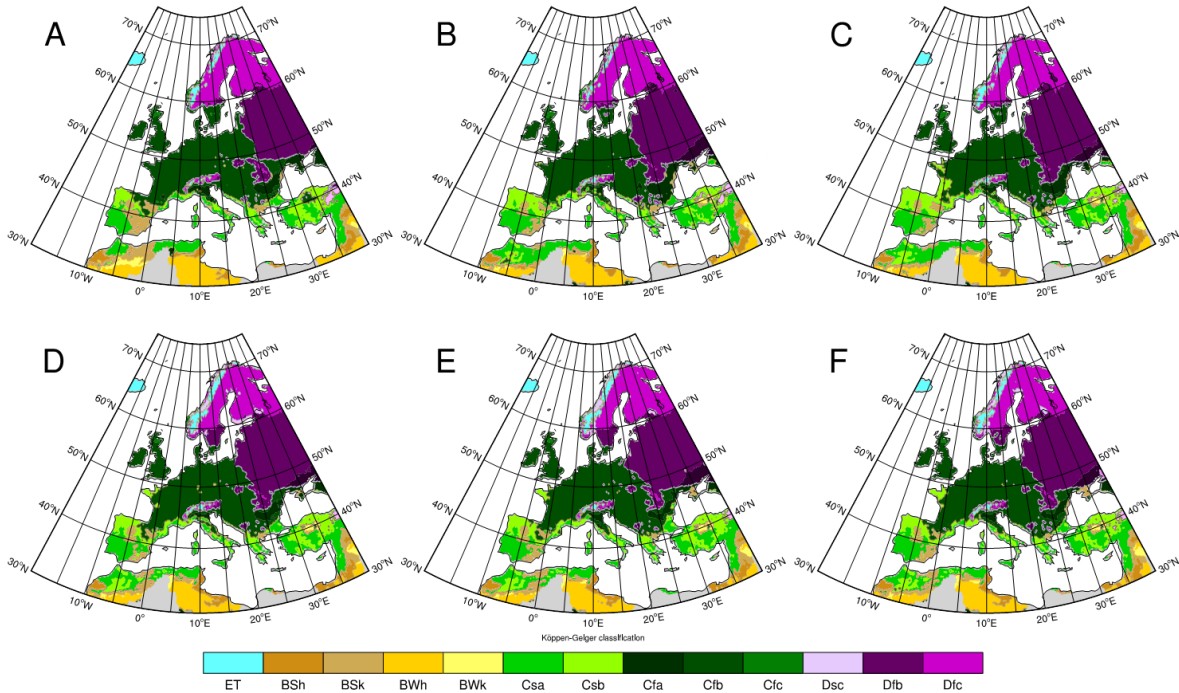

**Figure 9.** Simulated K-G climate classification according to E-OBS (**A**) and power transformation of precipitation and variance scaling of temperature correction in ALADIN (**B**), HIRHAM (**C**), RegCM (**D**), RAMCMO2 (**E**) and RCA (**F**) in the test period in 1981−2000.

Even though eQM produced a very small residual bias in ALADIN and RegCM in the Mediterranean and in RACMO2 in Eastern-Europe, the power transformation of precipitation better reproduced the K-G zone distribution. Furthermore, the power transformation of precipitation resulted in a smaller residual bias in Scandinavia, France and in the British Isles in both seasons, and in Mid-Europe in winter compared to eQM (Table 9a). The difference between the eQM and the variance scaling corrected temperature was negligible (Table 9b).

**Table 9.** Residual bias of seasonal amount of simulated precipitation (a) in the case of power transformation of precipitation and the seasonal mean of the simulated temperature, (b) in the case of variance scaling of the temperature bias correction methods in the test period in eight different regions: the Alps (AL), the British Isles (BI), Eastern Europe (EA), France (FR), the Iberia Peninsula (IP), the Mediterranean (MD), Mid-Europe (ME) and Scandinavia (SC) in DJF and JJA. The bias values are in % and in °C for precipitation and temperature, respectively.

| a) | ALADIN | | HIRHAM | | RegCM | | RACMO2 | | RCA | |
|---|---|---|---|---|---|---|---|---|---|---|
| Region | DJF | JJA | DJF | JJA | DJF | JJA | DJF | JJA | DJF | JJA |
| AL | 1 | 16 | 4 | −7 | 7 | −2 | 8 | 7 | 6 | 5 |
| BI | −8 | 9 | −12 | 9 | −12 | −5 | −15 | −6 | −15 | −9 |
| EA | 5 | 20 | 4 | 11 | 11 | 8 | 11 | 8 | 14 | 9 |
| FR | −7 | 10 | −7 | −9 | −11 | −12 | −11 | −14 | −11 | −11 |
| IP | 15 | 12 | 14 | 123 | 25 | 11 | 26 | 5 | 25 | −4 |
| MD | 7 | 3 | 8 | 23 | 20 | 7 | 24 | 20 | 20 | 15 |
| ME | −3 | 23 | −4 | 8 | −3 | 4 | −8 | 3 | −2 | 3 |
| SC | −12 | 1 | −12 | 1 | −11 | −6 | −8 | −9 | −9 | −8 |

| −100 | −50 | −25 | −10 | 10 | 25 | 50 | 100 |

**Table 9.** *Cont.*

| b) | ALADIN | | HIRHAM | | RegCM | | RACMO2 | | RCA | |
|---|---|---|---|---|---|---|---|---|---|---|
| Region | DJF | JJA | DJF | JJA | DJF | JJA | DJF | JJA | DJF | JJA |
| AL | 0.1 | −0.8 | 0.2 | −0.8 | −0.5 | −0.4 | −0.7 | −0.6 | −0.6 | −0.6 |
| BI | 0.4 | −0.6 | 0.2 | −0.7 | −0.6 | −0.3 | −0.6 | −0.2 | −0.6 | −0.4 |
| EA | −0.2 | −0.1 | −0.1 | −0.1 | −0.2 | −0.5 | −0.3 | −0.5 | −0.2 | −0.6 |
| FR | 0.1 | −1.1 | 0.1 | −1.1 | −0.7 | −0.4 | −0.9 | −0.4 | −0.9 | −0.5 |
| IP | −0.3 | −0.5 | −0.2 | −0.6 | −0.6 | 0.1 | −0.6 | 0.1 | −0.7 | 0.0 |
| MD | 0.1 | −0.1 | 0.2 | −0.4 | 0.0 | −0.5 | −0.1 | −0.8 | −0.2 | −0.7 |
| ME | −0.1 | −0.6 | 0.1 | −0.6 | −0.4 | −0.4 | −0.5 | −0.5 | −0.4 | −0.6 |
| SC | 0.8 | 0.4 | 0.9 | 0.0 | −0.2 | −0.2 | −0.2 | −0.3 | 0.0 | −0.2 |

| 5 | 2 | 1 | 0.5 | −0.5 | −1 | −2 | −5 |
|---|---|---|---|---|---|---|---|

### 3.5.4. Validation of Local Intensity Scaling of Precipitation

The local intensity scaling of precipitation was also combined with the variance scaled of temperature to calculate the K-G zone. This combination produced a similar K-G distribution as the combination of power transformation of precipitation and variance scaling of temperature in some regions except in the HIRHAM RCM (Figure 10). Based on the LOCI bias correction, the Csb climate zone occurred in Eastern Europe, and the BSk zone decreased in the Eastern region of the Mediterranean. The extension of the Dsc zone in Scandinavia and the extension of the Csb zone in Western France decreased in the ECHAM5-r3 driven RCMs compared to the power transformation variance scaling bias correction combination.

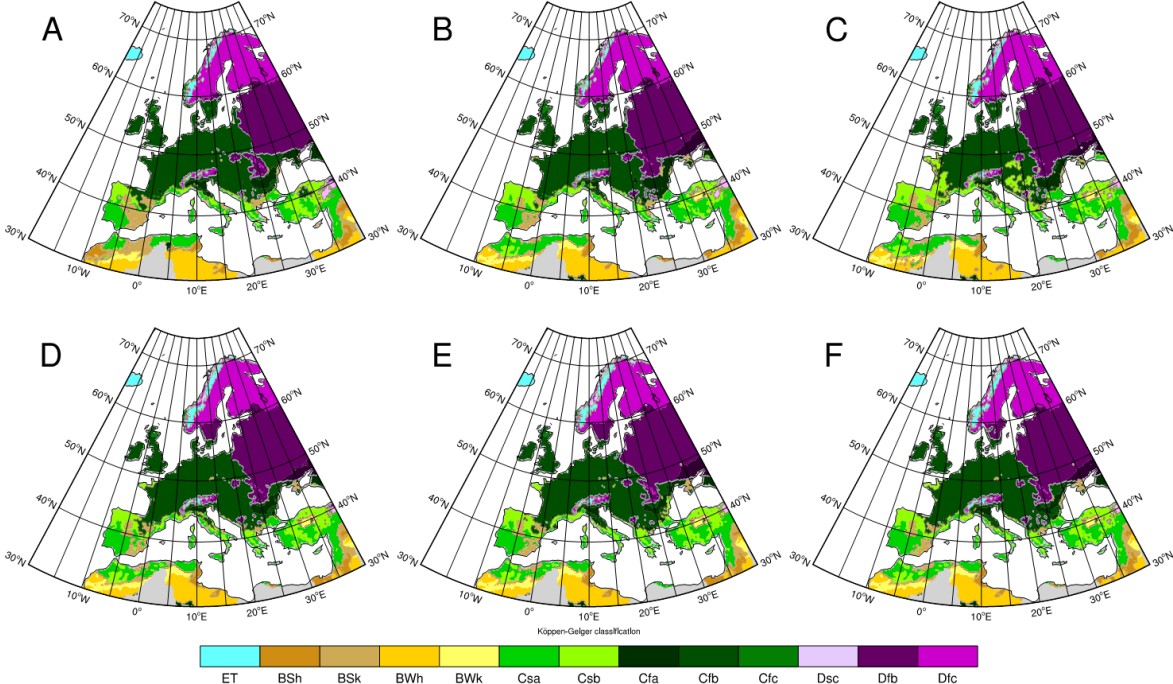

**Figure 10.** Simulated K-G climate classification according to E-OBS (**A**) and local intensity scaling of precipitation and variance scaling of temperature correction in ALADIN (**B**), HIRHAM (**C**), RegCM (**D**), RAMCMO2 (**E**) and RCA (**F**) in the test period in 1981−2000.

The residual bias of LOCI was extremely large in the HIRHAM model in the Iberian Peninsula in summer (Table 10). The residual bias increased in France compared to power transformation of the precipitation method. In contrast to this, the residual bias was smaller in Eastern Europe in the ECHAM5-r3 forced RCMs.

**Table 10.** Residual bias of the seasonal amount of simulated precipitation (a) in the case of local intensity scaling of the precipitation bias correction method in the test period in eight different regions: the Alps (AL), the British Isles (BI), Eastern Europe (EA), France (FR), the Iberia Peninsula (IP), the Mediterranean (MD), Mid-Europe (ME) and Scandinavia (SC) in DJF and JJA. The bias values are in %. The extremely large bias in the case of HIRHAM RCM in IP in the in JJA season is denoted by NA where the bias value is about $3 \times 10^{12}$.

| | ALADIN | | HIRHAM | | RegCM | | RACMO2 | | RCA | |
|---|---|---|---|---|---|---|---|---|---|---|
| Regio | DJF | JJA | DJF | JJA | DJF | JJA | DJF | JJA | DJF | JJA |
| AL | 0 | 18 | 5 | −3 | 7 | −2 | 8 | 8 | 8 | 4 |
| BI | −10 | 8 | −14 | 9 | −14 | −7 | −17 | −9 | −17 | −12 |
| EA | −1 | 19 | −3 | 11 | 4 | 6 | 4 | 7 | 7 | 6 |
| FR | −11 | 8 | −11 | −6 | −15 | −16 | −16 | −18 | −16 | −16 |
| IP | 13 | 14 | 11 | NA | 24 | 12 | 24 | 4 | 22 | −4 |
| MD | 4 | 5 | 6 | 43 | 19 | 5 | 23 | 27 | 18 | 14 |
| ME | −9 | 22 | −9 | 6 | −9 | 2 | −14 | 0 | −8 | 0 |
| SC | −16 | 0 | −16 | −1 | −15 | −8 | −13 | −11 | −14 | −10 |

−100 −50 −25 −10 10 25 50 100

## 4. Discussion

The results confirmed our supposition that bias corrections have a significant effect on climate classification. The effect of the bias correction varied among the models and the regions of the model domains. Table 11 shows the differences between the observed and simulated Köppen−Geiger climate zones in each region. The results were received by the calculation of the number of grid points where the simulated and observed K-G zones were different in each region, and then this number was divided with the number of grid points of the regions. The eQM and gpQM resulted in the largest differences between the RCMs. These differences stemmed from the correction of precipitation. Simulated precipitation is very sensitive to the properties of a model, e.g., physical parameterization, surface properties, and resolution; hence, the distribution of precipitation varied among the RCMs. In the HIRHAM model, eQM and gpQM produced drier negative bias, i.e., dryer zones in almost the entire studied area. The dominance of these dry climate classes originated from the surface properties in the HIRHAM model, the 1 mm threshold value of precipitation and the correction method. Unlike the other RMCs, HIRHAM has only one soil moisture layer [50], which results in a smaller water-holding capacity, which probably causes a negative feedback effect on precipitation formation. Owing to the threshold value, most of the daily mean precipitation values were less than 1 mm, which were resized to zero. This threshold value also caused negative precipitation bias in the JJA season. Moreover, the eQM corrected the ranked category, but not the value of the variable. Hence, the precipitation (or temperature) values transformed into "very high" values correspond to what observations tell us about actual "very high "values [15]. Notwithstanding that the eQM is expected to be the best method according to some literature [15,51,52], but according to some studies, the distribution-based methods improve the RCMs [31,44,53]. The remaining large biases may originate from the weakness of linear extrapolation of the cumulative distribution of parameters.

**Table 11.** Disagreement between observed and simulated K-G zones in eight different regions: the Alps (AL), the British Isles (BI), Eastern Europe (EA), France (FR), the Iberia Peninsula (IP), the Mediterranean (MD), Mid-Europe (ME) and Scandinavia (SC) and in the whole study area in DJF and JJA in the case of eQM-eQM, gpQM-eQM, power transformation of precipitation and variance scaling of temperature and LOCI and variance scaling of temperature bias correction combination. The values are in %.

| DISAGREEMENT | AL | BI | EA | FR | IP | MD | ME | SC | Study Area |
|---|---|---|---|---|---|---|---|---|---|
| **eQM-eQM** | | | | | | | | | |
| ALADIN | 8.9 | 1.5 | 15.5 | 8.8 | 12 | 17.8 | 2.3 | 4.5 | 9 |
| HIRHAM | 38.6 | 1.5 | 29.5 | 44.2 | 31.6 | 35.4 | 2.4 | 7.8 | 20 |
| RegCM | 9.4 | 1.3 | 10.3 | 1.5 | 16.2 | 17.2 | 0.7 | 6 | 8.4 |
| RACMO2 | 8.9 | 2.2 | 12.9 | 1.5 | 12.9 | 19 | 0.7 | 6.4 | 9 |
| RCA | 10.3 | 2.5 | 8.6 | 1 | 14.2 | 11.7 | 0.9 | 10.5 | 9 |
| Ensemble mean | 15.2 | 1.8 | 15.4 | 11.4 | 17.4 | 20.2 | 1.4 | 7.0 | |
| **gpQM-eQM** | | | | | | | | | |
| ALADIN | 11.8 | 2.7 | 18.5 | 39.3 | 19.3 | 29.9 | 3.4 | 4.7 | 13.1 |
| HIRHAM | 73.9 | 46.9 | 47.4 | 99 | 51.2 | 55.8 | 46.9 | 8.9 | 38.2 |
| RegCM | 9.9 | 1.5 | 10.9 | 2.5 | 17.4 | 21.8 | 0.5 | 6.6 | 9.4 |
| RACMO2 | 24.7 | 2.4 | 17.5 | 6.4 | 19.5 | 36.8 | 1.5 | 7.2 | 13.7 |
| RCA | 18.1 | 2.5 | 10 | 4.7 | 19.5 | 20.1 | 0.9 | 11.2 | 11.4 |
| Ensemble mean | 27.7 | 11.2 | 20.9 | 30.4 | 25.4 | 32.9 | 10.6 | 7.7 | |
| **power_variance** | | | | | | | | | |
| ALADIN | 1.2 | 0 | 0 | 0 | 2.9 | 3.8 | 0 | 0.2 | 0.8 |
| HIRHAM | 0 | 0 | 0 | 0 | 0 | 0 | 0 | 0.2 | 0.1 |
| RegCM | 0 | 0 | 0 | 0 | 0 | 0 | 0 | 0.2 | 0.1 |
| RACMO2 | 0 | 0 | 0 | 0 | 0 | 0 | 0 | 0.2 | 0.1 |
| RCA | 0 | 0 | 0 | 0 | 1.1 | 0 | 0 | 0.2 | 0.2 |
| Ensemble mean | 0.2 | 0.0 | 0.0 | 0.0 | 0.8 | 0.8 | 0.0 | 0.2 | |
| **loci-variance** | | | | | | | | | |
| ALADIN | 0 | 0 | 0.3 | 1.2 | 2.7 | 1.1 | 0.8 | 0.3 | 0.7 |
| HIRHAM | 0 | 0 | 0.4 | 2.5 | 3.2 | 1.4 | 0.8 | 0.3 | 0.8 |
| RegCM | 0 | 0 | 0.3 | 1.2 | 2.7 | 1.1 | 0.8 | 0.3 | 0.7 |
| RACMO2 | 0 | 0 | 0.3 | 1.2 | 2.9 | 3.4 | 0.8 | 0.3 | 0.9 |
| RCA | 0 | 0 | 0.3 | 1.2 | 2.6 | 1.2 | 0.8 | 0.3 | 0.7 |
| Ensemble mean | 0 | 0 | 0.3 | 1.4 | 2.8 | 1.6 | 0.8 | 0.3 | |

| 100 | 50 | 30 | 15 | 5 |
|---|---|---|---|---|

The model results corrected by the gpQM resulted in a similar climate classification to the eQM corrected simulations, regardless of the gpQM using gamma and generalized Pareto distributions. The remained bias can be explained by the fact that daily precipitation cannot be adequately expressed by gamma distribution for every region of Europe [54].

The power transformation of precipitation and the local intensity scaling of precipitation combined with the variance scaling of temperature performed correct K-G zone distribution with a negligible difference from the observed one. Furthermore, they resulted in very similar values in each of the RCMs. Their independence on the model and regions of the model domain can be explained by the fact that these are distribution-free correction approaches. Furthermore, they are also able to adjust the variance statistics of the precipitation time series, the simulated wet-day intensity, the wet-day frequency of precipitation and the variance and the mean values of temperature.

The bias correction methods were validated through a split-sample test by calculating the K-G zones in the 1981−2000 time period, except for the local intensity scaling of precipitation. According to the climate classification, the power transformation of precipitation and the variance scaling of temperature combination performed best in terms of K-G zones, despite the fact that the eQM bias correction methods had a smaller residual bias value in some RCMs, e.g., in ALADIN in the JJA season.

The bias correction methods were tested by the differential split-sample test in [44]. According to the statistical evaluation of the bias corrections in the test period, they found that the best method was distribution mapping based on gamma distribution, which was able to correct statistical moments other than means and standard deviations. Their findings presumably stemmed from their decision to choose smaller sized domains, in which only one European region was taken into account. We found the eQM and gpQM of precipitation had great limitations in the larger sized pan-European domain and produced incorrect climate classification in each RCM.

## 5. Conclusions

In this paper, the influence of bias corrections on K-G climate classification was investigated. Climate classification was calculated by eQM-corrected precipitation and temperature, by a combination of gpQM-corrected precipitation and eQM of temperature, by a combination of power transformation of precipitation and variance scaling of temperature, or by a combination of LOCI for precipitation and variance scaling for temperature. These bias correction methods were applied in five 25 km resolution ENSEMBLE RCMs in the historical time period of 1961−1990 and their results were compared with climate classification based on E-OBS-observed precipitation and temperature values to study their performance. The corrections were tested by a split-sample test, where the 1961−1980 period was training, and the corrections were validated in the 1981−2000 period. Subsequently, the climate classification was evaluated in eight individual subdomains: the Alps, the British Isles, Eastern Europe, France, the Iberian Peninsula, the Mediterranean, Mid-Europe and Scandinavia, defined according to the methodology devised for the PRUDENCE project.

When assessing the performance of the bias correction methods, we found similar results for eQM- and gpQM-corrected K-G classifications when daily data were used during the whole 30-year time period (not shown). Both of them were strongly dependent on the RCM, as the simulated climate zones varied between these RCMs. Moreover, the simulated climate zones significantly differed from the observed ones. These differences stemmed from the large bias in the seasonal precipitation amount. The 90-day moving window improved these correction methods. In comparison, a combination of LOCI and power transformation for precipitation with variance scaling of temperature, respectively, properly reproduced the climate zones by each of the RCMs in each region in the historical period. Furthermore, their test run contained the smallest differences from the observed K-G zones in most regions.

Our results suggest that the eQM and gpQM methods manifest a strong dependence on the spatial distribution of parameters, and this dependence causes a limitation in climate classification considering the large domain. Conversely, power transformation–and local intensity scaling of precipitation and variance scaling of temperature corrections–also generated a smaller bias between the simulated and observed parameters, except in HIRHAM in JJA, but their combination produced better results in climate classification for the whole European domain. This can be explained by the fact that they are distribution-free approaches.

This study is valid for Europe as a whole, since it was based on the E-OBS dataset with a resolution that may be coarser than that of some small regions studied in the quoted papers, where dense national datasets could be used. In the latter case, the statistical properties of the points reflect the smaller area and the results of the method evaluations could be different. It was beyond the scope of this study to devote itself to the several high-resolution gridded datasets that exist in Europe, but this will be the topic of future investigation using the next generation EURO-CORDEX regional climate model simulations.

**Author Contributions:** Writing–original draft preparation, B.S.-T.; review and editing, P.S., A.F. and J.M.

**Funding:** This research was funded by Ministry of Education, Youth and Sports of CR within the National Sustainability Program I (NPU I), grant number LO1415.

**Acknowledgments:** We are thankful for the E-OBS data set from the EU-FP6 project ENSEMBLES (http://www.ensembles-eu.org) and the data provided by the ECA&D project (http://www.eca.knmi.nl).We would like to thank Sixto Herrera García for his help in bias correction and MeteoLab application. We also thank the anonymous reviewer(s) for their comments.

**Conflicts of Interest:** The authors declare no conflict of interest. The funders had no role in the design of the study; in the collection, analyses, or interpretation of data; in the writing of the manuscript, or in the decision to publish the results.

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
