# Peer review of "Influence of Bias Correction Methods on Simulated Köppen−Geiger Climate Zones in Europe"

_climate, doi:10.3390/cli7020018_

Round 1
Reviewer 1 Report
The authors evaluate five bias correction/adjustment methods applied on the five ENSEMBLES regional climate models wrt. EOBS gridded observational dataset.
The paper is well structured and easy to follow. Most of the methodological aspects are clearly presented and discussed. I'm happy to suggest this paper for the publication in the journal Climate after some minor corrections and adjustments are provided.
Minor comments:
ln 121: RESOULTION > RESOLUTION
ln 249 "inside the 95 percentile"; please clarify.
ln 328 "in both RCMs"; please clarify which two RCMs are you referring to.
ln 428 "The NA is about 3x10^12"; please clarify.
Minor to moderate comments:
the list of references is not consistently formatted. Please put some effort.
it is not clear which specific version of E-OBS dataset is applied in your study.
in Introduction, please comment on the existence of other climate classification schemes.
in Conclusions, please comment on the next generation of regional climate models' simulations, EURO-CORDEX.
Author Response
Response to Reviewer 1 Comments
Point 1: ln 121: RESOULTION > RESOLUTION
Response 1: It is excepted and corrected.
Point 2: ln 249 "inside the 95 percentile"; please clarify.
Response 2: It is clarified by the sentence: „Owing to the fact the seasonal temperature probability distribution does not fit to Gaussian distribution, due to non-Gaussian tails occurrence, the gpQM bias correction with a 90-day moving window was implemented only on the precipitation data.”
Point 3: ln 328 "in both RCMs"; please clarify which two RCMs are you referring to.
Response 3: It was a typo. It is corrected with „each RCMs”.
Point 4: ln 428 "The NA is about 3x10^12"; please clarify.
Response 4.: It is clarified by: „The extremely large bias in the case of HIRHAM RCM in IP in the in JJA season is denoted by NA where the bias value is about 3x1012.”
Point 5: the list of references is not consistently formatted. Please put some effort
Response 5. It is corrected.
Point 6: it is not clear which specific version of E-OBS dataset is applied in your study.
Response 6: E-OBS version 10.0 was used. It is added in the manuscript too.
Point 7: in Introduction, please comment on the existence of other climate classification schemes
Response 7: The Köppen and Köppen-Trewartha classification schemes have already commented on.
Point 8: in Conclusions, please comment on the next generation of regional climate models' simulations, EURO-CORDEX.
Response 8: We added to the conclusion that we used ENSEMBLE RCMs but we will use EURO-CORDEX next generation simulations in the future.
Reviewer 2 Report
This is a solid paper focusing on bias correction methods on European climate zones. The article is well structured and it defines clearly presented key questions for itself. The approach and methodology is clearly presented and the quality of cartography is good. On page 13 the b part of Table 5 should be discricarded as it presents only 0 values. Additionally, the large number of highly similar maps cause difficulties in perceiving their differences. It is recommended that the authors could think about other ways of presenting their key results as the differences are very small. This is also recognized in the text itself (dicussion). The paper concludes with interpretations deriviable from the research design. Overall, this is publishable work.
Author Response
Response to Rewiever 2 comments
Point 1: On page 13 the b part of Table 5 should be discricarded as it presents only 0 values
Response 1: It is accepted and the Table 5 b is discricarded.
Point 2: . Additionally, the large number of highly similar maps cause difficulties in perceiving their differences. It is recommended that the authors could think about other ways of presenting their key results as the differences are very small. This is also recognized in the text itself (dicussion).
Response 2: The frequency of occurrence of the climate zones in each region was also calculated and the results were used to present the key results as the differences are very small according to some bias corrections. Moreover, the differences between the observed and simulated zones in each region were also presented with a table in the discussion.